# MicroRNA-214-3p Targeting Ctnnb1 Promotes 3T3-L1 Preadipocyte Differentiation by Interfering with the Wnt/β-Catenin Signaling Pathway

**DOI:** 10.3390/ijms20081816

**Published:** 2019-04-12

**Authors:** Feng-xue Xi, Chang-sheng Wei, Yan-ting Xu, Lu Ma, Yu-lin He, Xin-e Shi, Gong-she Yang, Tai-yong Yu

**Affiliations:** 1Key Laboratory of Animal Genetics, Breeding and Reproduction of Shaanxi Province, Yangling, Xianyang 712100, China; xifengxue627@163.com (F.-x.X.); weichshm@163.com (C.-s.W.); 1379356497@nwafu.edu.cn (Y.-t.X.); malu950712@nwafu.edu.cn (L.M.); 18302977505@163.com (Y.-l.H.); xineshi@163.com (X.-e.S.); gsyang999@hotmail.com (G.-s.Y.); 2Laboratory of Animal Fat Deposition & Muscle Development, College of Animal Science and Technology, Northwest A&F University, Yangling, Xianyang 712100, China

**Keywords:** miR-214-3p, proliferation, differentiation, Ctnnb1, adipogenesis

## Abstract

Differentiation from preadipocytes into mature adipocytes is a complex biological process in which miRNAs play an important role. Previous studies showed that miR-214-3p facilitates adipocyte differentiation of bone marrow-derived mesenchymal stem cells (BMSCs) in vitro. The detailed function and molecular mechanism of miR-214-3p in adipocyte development is unclear. In this study, the 3T3-L1 cell line was used to analyze the function of miR-214-3p in vitro. Using 5-Ethynyl-2′-deoxyuridine (EdU) staining and the CCK-8 assay, we observed that transfection with the miR-214-3p agomir visibly promoted proliferation of 3T3-L1 preadipocytes by up-regulating the expression of cell cycle-related genes. Interestingly, overexpression of miR-214-3p promoted 3T3-L1 preadipocyte differentiation and up-regulated the expression of key genes for lipogenesis: PPARγ, FABP4, and Adiponectin. Conversely, inhibition of miR-214-3p repressed 3T3-L1 preadipocyte proliferation and differentiation, and down-regulated the expression of cell cycle-related genes and adipogenic markers. Furthermore, we proved that miR-214-3p regulates 3T3-L1 preadipocyte differentiation by directly targeting the 3′-untranslated regions (3′UTR) of Ctnnb1, which is an important transcriptional regulatory factor of the Wnt/β-Catenin pathway. Taken together, the data indicate that miR-214-3p may positively regulate preadipocyte proliferation and enhance differentiation through the Wnt/β-Catenin signaling pathway.

## 1. Introduction

White adipose tissue is a specialized organ that stores fat and releases lipid droplets in response to various signals [1]. Meanwhile, an increasing number of studies have shown that adipose tissue also regulates metabolic homeostasis by secreting adipokines [2,3]. However, excessive accumulation of adipose tissue can cause metabolic disorders and lead to obesity [4]. Obesity is currently one of the most significant health issues in the world, and the problem is getting worse. Until 2016, 39% of adults 18 years and older were overweight worldwide [5]. Obesity is frequently associated with comorbidities, including diabetes, cardiovascular diseases, hyperglycemia, nonalcoholic fatty liver disease (NAFLD), cancer, and osteoarthritis [6,7,8,9]. In terms of a series of serious health problems, it is urgent to develop new and effective strategies to control and treat obesity. The main cause of corpulence is hypertrophy and hyperplasia of adipocytes, which can lead to an increase in body fat and metabolic disorders. Therefore, a detailed understanding of the mechanism of adipogenesis is important for the prevention and treatment of obesity and related metabolic disease.

MicroRNAs (miRNAs) are a class of endogenous non-coding RNAs with a length of about 22 nt, which are widely involved in post-transcriptional regulation activities via binding to the 3’UTR of their drones; most of them are highly sequence-conserved, expression-sequenced, and tissue-specific [10,11,12]. A growing body of research has shown that miRNAs play vital roles in adipogenesis. For example, Li et al. showed that miR-103 promotes 3T3-L1 cells adipogenesis by targeting MEF2D [13]; Pen et al. reported that miR-429 suppresses differentiation and promotes proliferation in porcine preadipocytes [14]; another study proved that miR-143-3p positively regulates preadipocyte differentiation by targeting MAPK7 [15]. However, the impact of novel miRNAs on the function and mechanism of adipogenesis still requires further research.

miR-214 is located in the Dnm3os transcribed locus [16] and is produced from Dnm3os RNA [17,18], which is highly conserved among its seed sequence. Many studies suggested that miR-214 plays important roles in different biological processes, such as bone formation, skeletal muscle development, hematopoiesis, and cancer [19,20,21,22,23]. Previously, several studies showed that the expression of miR-214 was significantly down-regulated in 3T3-L1 preadipocytes treated with LiCl [24]. In addition, expression of miR-214 was up-regulated after adipogenic differentiation in mouse embryonic stem cells [25]. These discoveries imply that miR-214-3p may play a crucial role in adipogenesis. Nevertheless, the definite function of miR-214-3p in adipogenesis is still unclear.

Here, we profiled the expression pattern of miR-214-3p during 3T3-L1 preadipocyte proliferation and differentiation. We found that miR-214-3p accelerated the proliferation and differentiation of 3T3-L1 preadipocytes by regulating cell cycle and adipogenic genes, respectively. These were partially mediated by targeting Ctnnb1 in adipogenesis. Overall, our study suggests that miR-214-3p is a positive regulator in 3T3-L1 preadipocyte proliferation and differentiation.

## 2. Results

### 2.1. The Expression of miR-214-3p Is Strongly Associated with Lipogenesis

To clarify the expression of miR-214-3p in various tissues of mice, total RNA was isolated from eight different tissues of adult C57BL/6J mice and real-time fluorescence quantification was performed which showed that miR-214-3p was highly expressed in white adipose tissue, brown adipose tissue, muscle, lung, and heart, but relatively lower expression was observed in the kidney, spleen, and liver (Figure 1A). This suggested a potential role of miR-214-3p in regulating adipogenesis. To determine the relationship between miR-214-3p and lipogenesis, C57BL/6J mice were randomly divided into two groups according to their weight and were fed a high-fat diet (HFD) or chow diet (CD) for 12 weeks. The HFD group showed a significant increase in body weight of 31.4% compared to the CD group (Figure 1B), and total triglycerides (TG) and cholesterol (TC) in serum were also increased (Figure 1C,D). In addition, the level of miR-214-3p was increased in the adipose tissue of HFD mice, especially in inguinal fat tissue (iWAT) (Figure 1E). These results indicated that miR-214-3p might be involved in adipogenesis.

### 2.2. The Profile of miR-214-3p in Adipogenesis

The mature sequence of miR-214-3p was highly conserved across multiple species, such as mouse, pig, human, and rat (Figure 2A). In an effort to study the role of miR-214-3p in adipocyte development, we examined the expression of miR-214-3p in 3T3-L1 preadipocytes during the proliferation and differentiation stages. We found that miR-214-3p first showed an increasing trend and then decreased during proliferation and differentiation (Figure 2B,C). We further performed Gene Ontology (GO) analysis on the target of miR-214-3p and found that miR-214-3p may be involved in adipocyte development, especially cell proliferation and differentiation (Figure 2D). Moreover, the Kyoto Encyclopedia of Genes and Genomes (KEGG) pathway analysis showed that miR-214-3p regulated adipocyte biological processes via the Wnt signaling pathway (Figure 2E). Therefore, we inferred that miR-214-3p may have a potentially important role in the adipocyte development process.

### 2.3. MicroRNA-214-3p Promotes 3T3-L1 Preadipocyte Proliferation

Lipid deposition in adipose tissue is associated with the proliferation and differentiation of preadipocytes [26]. To confirm the function of miR-214-3p in adipogenesis, we detected whether miR-214-3p impacts the proliferation of 3T3-L1 preadipocytes after transfecting with the miR-214-3p agomir, antagomir, and the negative control. The result showed that the transfection with the agomir remarkably increased the miR-214-3p expression level (Figure 3A). The mRNA and protein level of cell cycle-related genes, including Cyclin B, Cyclin D, Cyclin E, and CDK4, increased (Figure 3B–D). The EdU staining assay showed that EdU positive cells increased in the miR-214-3p agomir group compared to the negative control group (Figure 3E,F). Furthermore, cell viability was enhanced according to the CCK-8 assay (Figure 3G). To further demonstrate the influence of miR-214-3p on adipocyte proliferation, we transfected preadipocytes with the miR-214-3p antagomir. We found that the miR-214-3p was dramatically reduced by 98.9% (Figure 4A). The qRT-PCR and Western blot data showed that inhibition of miR-214-3p depressed the expression of cell cycle genes (Figure 4B–D). In addition, the EdU staining assay certified that inhibition of miR-214-3p reduced the EdU positive cells (Figure 4E,F). CCK-8 analysis showed that the cell number decreased after transfection with the miR-214-3p antagomir (Figure 4G). These results implied that miR-214-3p positively regulates 3T3-L1 preadipocyte proliferation.

### 2.4. MicroRNA-214-3p Enhances 3T3-L1 Preadipocyte Differentiation

Mature adipocytes are an important part of white adipose tissue. Previously, Guo et al. showed that miR-214 promotes adipocyte differentiation of BMSCs [27]. To investigate the detailed function of miR-214-3p in adipocyte differentiation, we transfected agomir, antagomir, or the negative control into 3T3-L1 preadipocytes, at a final concentration of 50 nM. After transfection with agomir or antagomir, the miR-214-3p was remarkably increased or decreased at 0, 2, 4, and 6 days after adipogenic induction (Figure 5A and Figure 6A). By boron-dipyrromethene (BODIPY) staining, we observed that overexpression of miR-214-3p increased the BODIPY positive cells, and the adipogenic capacity of 3T3-L1 preadipocytes was visibly enhanced (Figure 5E,F). While inhibition of miR-214-3p decreased the BODIPY positive cells, the adipogenic ability of preadipocytes was distinctly reduced when compared with the negative control (Figure 6E,F). Furthermore, using qPCR and Western blot analysis, we found that miR-214-3p agomir visibly enhanced both mRNA and protein level of adipogenic markers, including PPARγ, FABP4, and adiponectin, at the sixth day of preadipocyte differentiation (Figure 5B–D), while they were impaired in the antagomir group (Figure 6B–D). In addition, it was also confirmed by Oil Red O staining that overexpression of miR-214-3p enhanced the adipogenic capacity of preadipocytes (Figure 5G,H), while inhibition of miR-214-3p attenuated the adipogenic capacity of preadipocytes (Figure 6G,H). Collectively, the data revealed that miR-214-3p promotes 3T3-L1 preadipocyte differentiation.

### 2.5. MicroRNA-214-3p Promotes 3T3-L1 Preadipocyte Differentiation by Targeting Ctnnb1 and Regulating the Canonical Wnt/β-Catenin Signaling Pathway

To understand the potential regulatory mechanism of adipogenesis by miR-214-3p, we used TargetScan 7.0 to predict the potential target of miR-214-3p (Figure 7A). Among hundreds of target genes, we selected Ctnnb1 as a candidate target gene for miR-214-3p. Ctnnb1 encodes catenin (cadherin-associated protein), beta-1, which is a key signaling molecule in the canonical Wnt signaling pathway that controls adipocyte differentiation. Therefore, it caught our attention. We constructed a dual luciferase reporter vector for wild-type Ctnnb1 3’UTR and mutant Ctnnb1 3’UTR (Figure 7B). A dual luciferase reporter assay showed that co-transfection of miR-214-3p agomir and Ctnnb1 3′UTR dual-luciferase reporter vector significantly inhibited the activity of the wild psiCHECK-2-Ctnnb1-3′UTR reporter, while the mutant reporter vector did not change (Figure 7C). In addition, to further demonstrate the association of miR-214-3p with Ctnnb1, we examined the expression of Ctnnb1 during adipogenic differentiation and found that Ctnnb1 showed an opposite trend with miR-214-3p, especially in the middle and late stages of differentiation (Figure 7D). Therefore, we examined the mRNA (Figure 7E) and protein levels (Figure 7F,G) of Ctnnb1 in the mid-differentiation stage and found a remarkable decrease in the treatment of pre-adipocytes with the miR-214-3p agomir. Taken together, these data showed that miR-214-3p promotes adipocyte differentiation through directly targeting Ctnnb1 and inhibiting the Wnt/β-catenin signaling pathway. 

## 3. Discussion

In this study, we found that miR-214-3p plays a crucial role in regulating 3T3-L1 preadipocyte proliferation and differentiation. In particular, miR-214-3p agomir facilitated the proliferation and differentiation, while its antagomir suppressed the process in 3T3-L1 preadipocytes. Taken together, our findings provided evidence to reveal that miR-214-3p may be an important positive regulator in adipogenic development.

In recent years, obesity has become a widespread issue that plagues human health. The main cause of obesity is energy intake and output imbalance, which causes excess energy to be converted into fat that accumulates in the body, thus resulting in obesity [28]. Studies have shown that the adipogenic process, including proliferation and differentiation, can be regulated by a range of regulatory factors [29]. As important non-coding RNAs, miRNAs are involved in the regulation of this process [30]. In our study, we found that miR-214-3p is highly expressed in adipose tissue induced by a high-fat diet in mice (Figure 1E). This is consistent with previous research [31]. Previous studies showed that miR-214 inhibits breast cancer cell proliferation and negatively regulates Wnt/β-catenin signaling in breast cancer [32], and miR-214-3p promotes the proliferation of osteosarcoma cells by targeting CADM1 [33], which indicates that miR-214 is indeed involved in the regulation of cell proliferation. To investigate the function of miR-214-3p in the proliferation of preadipocytes, the agomir and antagomir of miR-214-3p were transfected into 3T3-L1 preadipocytes. We found that miR-214-3p contributed to 3T3-L1 preadipocyte proliferation by up-regulating the mRNA and protein levels of Cyclin B, Cyclin D, Cyclin E, and CDK4 (Figure 3B–D). Meanwhile, in order to intuitively observe the effect of miR-214-3p on the proliferation of 3T3-L1 preadipocytes, we performed the EdU staining assay and found that miR-214-3p noteworthily increased the number of EdU positive cells (Figure 3E,F). Interestingly, in this study, we found that miR-214-3p not only promoted 3T3-L1 adipocyte proliferation but also enhanced adipocyte differentiation. Consistent with this, miR-214 has been reported to have the same regulation during myoblast development. Inhibition of miR-214 expression represses proliferation and differentiation of C2C12 myoblasts [34]. Feng et al. used flow cytometry to detect the cell cycle and found that inhibition of miR-214-3p significantly reduced the number of cells in the S phase. It is well known that the S phase is the DNA synthesis phase, and Cyclin E plays an important role in this period [35]. In our study, we found that after overexpression or interference with miR-214-3p, Cyclin E protein expression levels were significantly increased or decreased. This suggests that miR-214-3p is likely to affect the proliferation of S phase cells by regulating the expression of Cyclin E. 

MicroRNA-214 and miR-199a belong to the vertebrate-specific microRNA family and are located in the DNM3 gene, transcribed from the antisense strand DNM3os of DNM3, and approximately 8 kb in length. MicroRNA-214 is about 6 kb away from miR-199a, and their seed sequences are different [17,18], so they have different regulatory mechanisms and biological functions. It has been reported that miR-199a-3p had low expression levels in adipose tissue of HFD mice, which promoted the proliferation of 3T3-L1 adipocytes while inhibiting adipocyte differentiation by regulating stearoyl-CoA desaturase (SCD) [36]; miRNA-199a-5p accelerated porcine preadipocyte proliferation and attenuated its differentiation [37]. Previous studies on miR-214-3p mostly focused on cancer, osteogenesis, and muscle development [19,20,21,22,23]. In recent years, there have been few studies on the association between miR-214 and fat development. Guo et al. reported that the overexpression of miR-214 effectively promoted the adipocyte differentiation of BMSCs [27]. After treatment of 3T3-L1 preadipocytes with lithium, the expression of miR-214 was significantly down-regulated after induction of differentiation [24]. Nevertheless, the function and mechanism of miR-214-3p in adipogenic development is still unclear. In our study, we observed that miR-214-3p had high expression in mouse adipose tissue and during 3T3-L1 preadipocyte differentiation (Figure 1A and Figure 2C). Meanwhile, overexpression or inhibition of miR-214-3p enhanced or reduced the mRNA level of the adipogenic marker gene (Figure 5B and Figure 6B). Similarly, PPARγ, Adiponectin, and aP2 protein levels were also elevated or weakened compared to the control group (Figure 5C and Figure 6C), indicating that miR-214-3p positively regulates adipogenesis. Brown adipocytes consume energy, whereas white adipocytes store energy. Beige adipocytes are non-classical brown adipocytes present in white adipose tissue that are produced after exposure to a cold stimulation signal and have a similar function as brown adipocytes [38]. He et al. reported that miR-214 inhibits the differentiation of brown and beige adipocytes to negatively regulate energy consumption [31]. Consistent with this, our results showed that miR-214-3p promotes adipogenic differentiation of 3T3-L1 cells to positively regulate energy storage. Although we found that miR-214-3p promotes the proliferation and differentiation of preadipocytes in the 3T3-L1 cell line, its precise regulation of adipocyte development requires specific knockouts in vivo for more in-depth investigation.

In general, miRNAs bind to the 3′UTR of target mRNAs to induce mRNA degradation or translational repression [10]. Therefore, we predicted that miR-214-3p may target Ctnnb1, which encodes the catenin beta-1 protein. β-catenin is the key mediator of canonical Wnt/β-catenin signaling and plays an important role in intercellular signaling and regulation [39,40,41]. Generally, the Wnt/β-catenin signaling pathway inhibits adipogenesis when it is activated [42]. It has been reported that the Wnt/β-catenin signaling pathway can control lipogenesis through miRNA. MicroRNA-135a-5p suppresses 3T3-L1 adipogenesis by activation of canonical Wnt/β-catenin signaling [43]. MicroRNA-344 represses 3T3-L1 cell differentiation via targeting the GSK3β of the Wnt/β-catenin signaling pathway [44]. So far, only a few studies have linked miR-214 to Wnt/β-catenin with cancer. Xu et al. reported that in esophageal cancer, miR-214 bound to 3′-UTR of β-catenin mRNA to inhibit its translation [45]. Chandrasekaran et al. showed that miR-214 suppressed the expression of Ctnnb1 in human cervical and colorectal cancer cells [46]. In our present study, we transfected the miR-214-3p agomir into 3T3-L1 preadipocytes and induced cell differentiation using induction medium. We found that miR-214-3p inhibits Ctnnb1 mRNA and protein expression (Figure 7E,F), suggesting that Ctnnb1 may be the target of miR-214-3p, thereby affecting the transcriptional and translational levels of Ctnnb1 and inhibiting the Wnt/β-catenin signaling pathway. Further, a dual-luciferase reporter assay showed that miR-214-3p significantly inhibits luciferase activity (Figure 7C), indicating that Ctnnb1 is a direct target of miR-214-3p in 3T3-L1 cells.

## 4. Materials and Methods

### 4.1. Experimental Animals

All animal experiments were conducted through strict observance of the U.K. Animals (Scientific Procedures) Act, 1986 (the Animal Care Commission of the College of Veterinary Medicine, Northwest A&F University (14-233, 10 December 2014). Eight-week-old male C57BL/6J mice were purchased from the animal center of Xi’an Jiaotong University and kept in a constant temperature environment. The mice were allowed to eat and drink freely. After a week of adaptation, the mice were randomly distributed into weight-matched groups and fed either a normal chow diet (CD; 15% fat) or a high-fat diet (HFD; 60% fat; TROPHIC Animal Feed High-Tech Co. LTD, Nantong, China). After 12 weeks of HFD feeding, animals were sacrificed and fat depots from inguinal white adipose tissue (iWAT), perigonadal visceral adipose tissues (vWAT), interscapular brown adipose tissue (BAT), and serum samples were rapidly removed, immediately frozen in liquid nitrogen, and stored at −80 °C until further analysis.

### 4.2. Serum Analysis

The blood sample was collected into anticoagulation tubes and immediately centrifuged for ten minutes with a 4 °C centrifuge (12,000 rpm). Then the upper serum was aspirated into new centrifuge tubes and stored at −80 °C until further analysis. Serum levels of triglycerides (TG) and cholesterol (TC) were determined (Servicebio technology CO., LTD, Wuhan, China).

### 4.3. Cell Culture and Differentiation

The 3T3-L1 cell line and Human embryonic kidney (HEK) 293T cell line were kindly provided by the Stem Cell Bank, Chinese Academy of Sciences. The cell lines were maintained in Dulbecco modified Eagle medium (DMEM, Gibco, CA, USA) supplemented with 10% fetal bovine serum (FBS, Gibco, USA) and antibiotics (100 U/mL penicillin and 100 μg/L streptomycin) at 37 °C in a 5% CO_2_, constant temperature and humidity atmosphere. When 3T3-L1 preadipocytes reached confluence two days later, adipogenic differentiation was carried out according to previously published protocols. The 3T3-L1 cells were stimulated for two days in differentiation medium: DMEM containing 10% FBS and MDI (0.5 mM IBMX, 1 μM DEX, and 5 μg/mL insulin). Then, cells were maintained in DMEM containing 10% FBS and 5 μg/mL insulin. The medium was replaced every two days.

### 4.4. Transfection of miRNA Agomir and Antagomir

An agomir is a type of specially labeled and chemically modified double-stranded microRNA, which can regulate the biological function of the target gene by mimicking endogenous microRNA. An antagomir is a type of specially labeled and chemically modified single-stranded microRNA, designed based on the mature microRNA sequence, which is special for inhibiting the expression of endogenous microRNA. The miR-214-3p agomir, antagomir, and respective nonspecific control (NC) were purchased from RiboBio (Guangzhou, China) and were transfected into 3T3-L1 cells by the X-tremeGENE HP DNA Transfection Reagent (Roche, Mannheim, Germany) at a final concentration of 50 nM according to the manufacturer’s protocol. After 24 h transfection, the medium was replaced by the growth medium DMEM.

### 4.5. RNA Isolation and RT-qPCR

Total RNA of cells and tissues was isolated using the RNAiso Plus (TaKaRa, Otsu, Japan) according to the manufacturer’s instructions. The synthesis of the cDNA was performed with reverse transcription kits (TaKaRa). Quantitative PCR analyses were performed using SYBR green (Vazyme, Nanjing, China) with the StepOne system (ABI, MA, USA). Primer sequences used for RT-qPCR are listed in Table 1.

### 4.6. Western Blot Analysis

Cells were cleaved on ice for 30 min by protein lysate (RIPA) (Beyotime, Shanghai, China) supplemented with a protease inhibitor (Pierce, Rockford, IL, USA). After centrifugation at 12,000 rpm for 10 min in a 4 °C centrifuge, the supernatant was aspirated into a 1.5 mL centrifuge tube. The total cellular protein was separated by 10% SDS–polyacrylamide gel and then transferred to a PVDF membrane (Millipore, Boston, MA, USA). Next, the membrane was sealed with 5% skim milk powder for 2 h. After this, the membrane was incubated with primary antibody overnight at 4 °C, followed by incubation of the secondary antibody for 1 h at room temperature. Protein bands were presented using chemiluminescence solutions and quantified using Image Lab and Document Image processing. Anti-PPARγ from Abcam (Cambridge, UK); anti-aP2, anti-Adiponectin, anti-Cyclin B, anti-Cyclin D, anti-Cyclin E, and anti-CDK4 from Santa Cruz (Dallas, TX, USA); and anti-β-actin and anti-β-catenin from Boster (Wuhan, China) were used.

### 4.7. EdU Staining

The 3T3-L1 cells were inoculated into 96-well plates for 24 h and then transfected with Cell-LightTM EdU Apollo 567 In Vitro Kit (RiboBio, Guangzhou, China) to detect DNA synthesis. Firstly, 100 μL of 50 μM EdU Reagent A was added per well (diluted 1:1000 with cell culture medium) and incubated for 2 h in the incubator. Cells were then fixed with 4% paraformaldehyde for 30 min at room temperature, neutralized with 2 mg/mL glycine for 5 min and penetrated for 5 min using 0.5% Trixon-100. According to the instructions, the cells were incubated with 1X Apollo (EdU reagent B, C, D, and E mixture) staining reaction solution for 30 min in the dark at room temperature. Then the cells were rinsed with 0.5% Trixon-100 three times and washed with methanol two times. Finally, the nucleus was stained with Hoechst for 30 min and washed three times with PBS. Images were captured with a fluorescence microscope (Nikon, Tokyo, Japan).

### 4.8. Cell Counting Kit

The 3T3-L1 cells were seeded in 96-well plates (2000 cells per dish). After transfecting the agomir or antagomir and negative control for 24 h, 10 μL of CCK-8 reagent (Solarbio, Beijing, China) was added to each well and incubated for 4 h in a 37 °C incubator. After that, the absorbance at 450 nm wavelength was measured.

### 4.9. Oil Red O staining

Adipocytes were washed three times with PBS and fixed with 4% paraformaldehyde for 30 min. After washing thrice with PBS, the adipocytes were stained with 1% filtered Oil Red O solution for 30 min. Then, adipocytes were washed and observed using an inverted microscope. For Oil Red O quantitative analysis, the intracellular adsorbed Oil Red O was extracted in 100% isopropanol, and absorbance was measured at 490 nm wavelength.

### 4.10. BODIPY Staining

Differentiated adipocytes of day six were washed three times with PBS and fixed with 4% paraformaldehyde for 30 min. Next, adipocytes were washed thrice with PBS and incubated with the lipophilic dye BODIPY (Invitrogen, CA, USA) (stock concentration 1 mg/mL, working concentration 1:1000 dilution) for 30 min. The adipocytes were washed three times with PBS and nuclei were stained with DAPI (Invitrogen, CA, USA) for 10 min. After washing with PBS, images were captured by a fluorescence microscope (Nikon, Japan).

### 4.11. Luciferase Reporter Assay

Luciferase reporter plasmids (psi-CHECK2) containing the wild-type 3′UTRs of Ctnnb1 (WT-Ctnnb1) and mutant 3′UTRs of Ctnnb1 (Mut-Ctnnb1) were manufactured by General Biosystems Co., Ltd. (Anhui, China). HEK293T was seeded in a 48-well, and X-tremeGENE HP DNA Transfection Reagent was used to co-transfect HEK293 T cells with the wild-type or mutant 3′UTR luciferase reporter plasmids, and the miR-214-3p agomir or the negative control, respectively. The cells were harvested 24 h after transfection, and the luciferase activities were measured using the Dual-Glo Luciferase Assay System (Promega; Madison, WI, USA), following the manufacturer’s instructions. Firefly luciferase was used as a normalization control.

### 4.12. Statistical Analysis

All results were presented as mean ± SEM. Statistical analyses were performed with GraphPad Prism 7 software (San Diego, CA, USA). At least three independent repetitions were conducted for each experiment. Comparison between two sets of samples was analyzed by Student’s t-test. Statistical significance was determined at * *p* < 0.05; ** *p* < 0.01, and *** *p* < 0.001.

## 5. Conclusions

To sum up, our study reveals that miR-214-3p is a novel regulator of 3T3-L1 preadipocyte development, which promotes 3T3-L1 preadipocyte proliferation and enhances 3T3-L1 preadipocyte differentiation through targeting the Wnt/β-catenin signaling pathway. These findings contribute to a better understanding of adipogenesis regulated by miRNA. Simultaneously, it also provides an important reference value for the study of the pathogenesis and treatment of obesity.

## Figures and Tables

**Figure 1 ijms-20-01816-f001:**
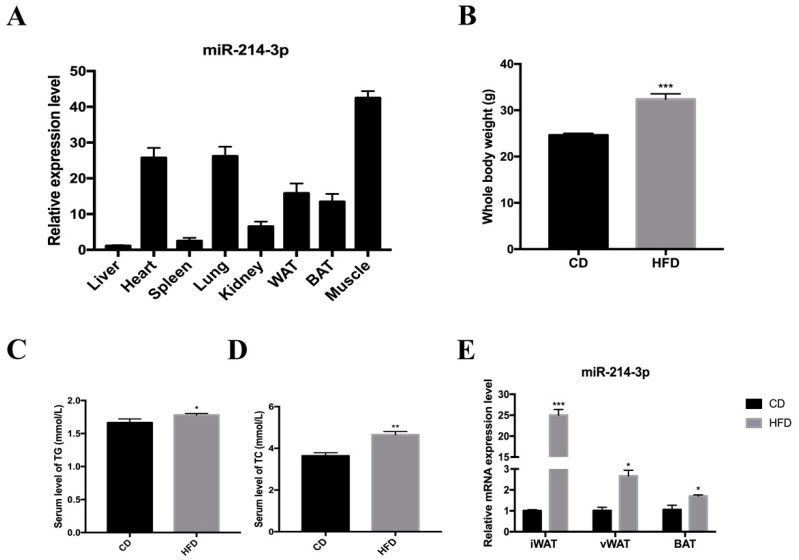
The expression of miR-214-3p is high in adipose tissue. (**A**) The relative expression level of miR-214-3p in eight different tissues of C57BL/6J mice vs. liver; (**B**) C57BL/6J mice were weighed after 12 weeks of being fed a high fat diet (HFD) or chow diet (CD); (**C**) the serum level of total triglyceride (TG); and (**D**) cholesterol (TC) were measured; (**E**) The expression of miR-214-3p in inguinal white adipose tissue (iWAT), perigonadal visceral adipose tissues (vWAT), and interscapular brown adipose tissue (BAT). Results were indicated as the mean ± SEM, *n* = 5. * *p* < 0.05; ** *p* < 0.01, *** *p* < 0.001.

**Figure 2 ijms-20-01816-f002:**
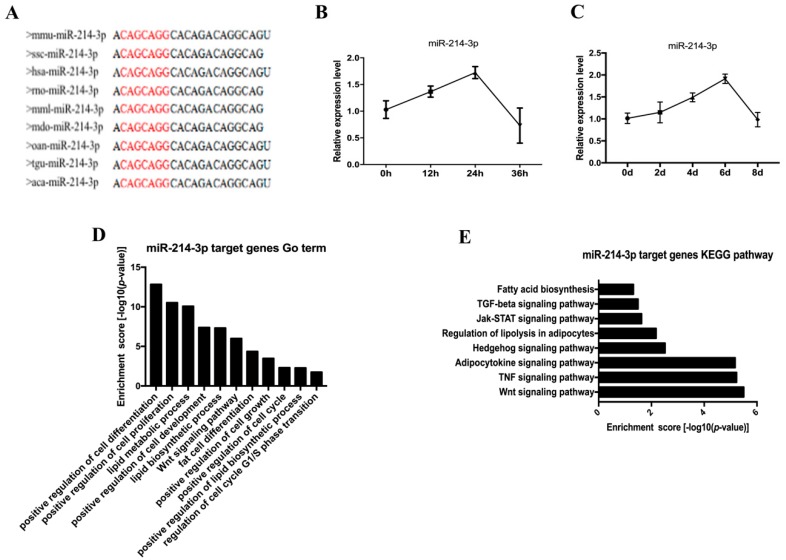
The level of miR-214-3p is up-regulated during 3T3-L1 adipogenesis. (**A**) The sequence of mature miR-214-3p is highly conserved across species; (**B**) RT-qPCR was performed to detect the expression of miR-214-3p during proliferation; (**C**) RT-qPCR analysis of miR-214-3p expression during adipogenic differentiation; (**D**) Go term analysis of the miR-214-3p targets; (**E**) KEGG pathway analysis of the miR-214-3p targets. Results were indicated as the mean ± SEM, *n* = 3. * *p* < 0.05; ** *p* < 0.01.

**Figure 3 ijms-20-01816-f003:**
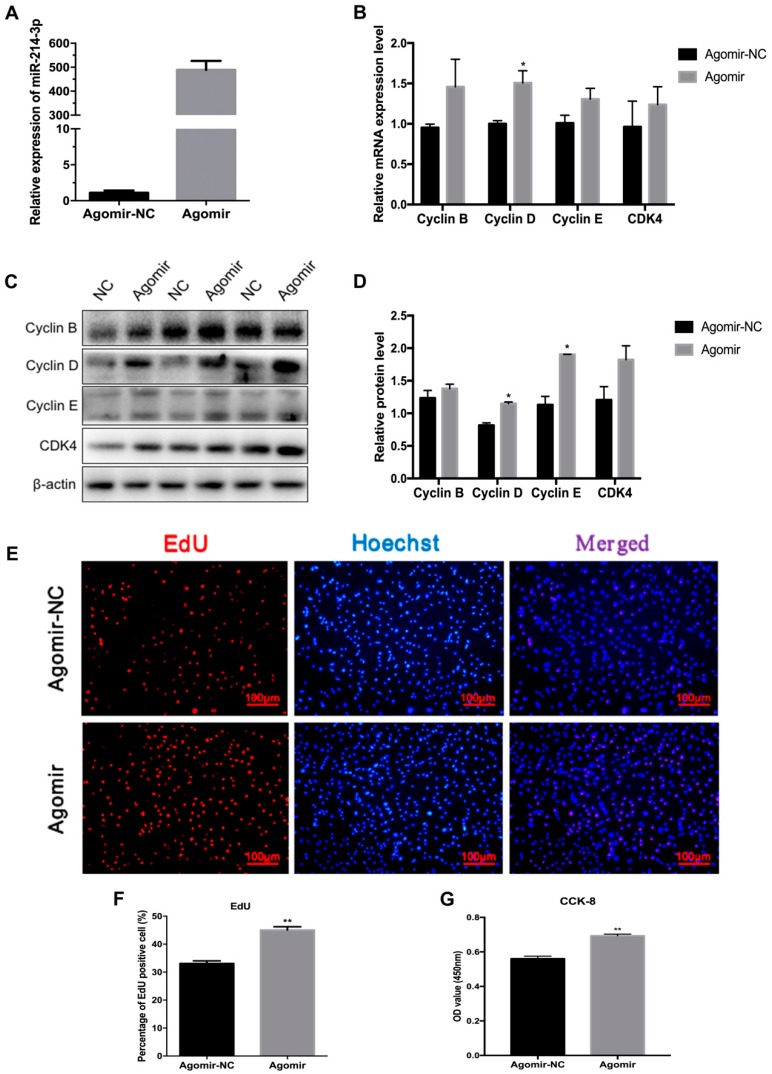
MiR-214-3p promotes the cell cycle process and upregulates the expression of cell cycle genes. MiR-214-3p agomir or negative control (NC) were transfected into cells at 50% density at 50 nM, and cells were harvested 24 h after transfection. (**A**) The overexpression efficiency of miR-214-3p after transfection with the miR-214-3p agomir compared with the negative control (NC); (**B**) Real-time qPCR was used to detect cell cycle genes, Cyclin B, Cyclin E, Cyclin D, and CDK4 after 24 h of transfection; (**C**) Western blot analysis of cell cycle genes; (**D**) Quantification of Western blot analysis of Cyclin B, Cyclin D, Cyclin E, and CDK4; (**E**) EdU staining assay was carried out after transfection for 24 h. Cells during DNA replication were stained by EdU (red), and cell nuclei were stained with Hoechst (blue), (bar size = 100 μm); (F) The percentage of EdU positive cells/Hoechst positive cells was quantified. (G) CCK-8 assay was used to detect cell viability after transfection for 24 h—results represent the absorbance value at 450 nm after incubation with 10% CCK-8 solution for 4 h. Statistical results were representative of mean ± SEM, *n* = 3. * *p* < 0.05; ** *p* < 0.01, vs. NC.

**Figure 4 ijms-20-01816-f004:**
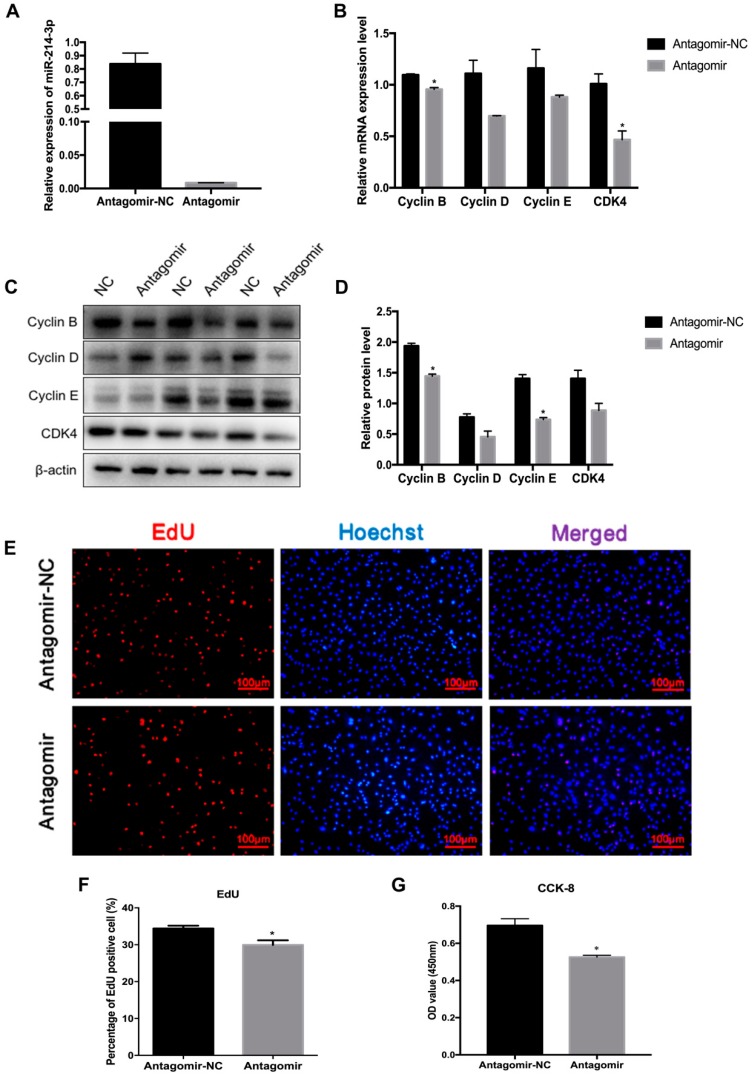
MiR-214-3p inhibitor inhibits the cell cycle process and down-regulates the expression of cell cycle genes. MiR-214-3p antagomir or negative control (NC) were transfected into cells at 50% density at 50 nM, and cells were harvested 24 h after transfection. (**A**) The knockdown efficiency of miR-214-3p after transfection with the miR-214-3p antagomir compared with the negative control; (**B**) RT-qPCR was used to detect the expression of cell cycle genes after transfection for 24 h; (**C**) Western blot analysis of Cyclin B, Cyclin E, Cyclin D, and CDK4; (**D**) Quantification of Western blot analysis of Cyclin B, Cyclin D, Cyclin E, and CDK4; (**E**) EdU staining of 3T3-L1 preadipocytes after transfection for 24 h. Cells during DNA replication were stained by EdU (red), and cell nuclei were stained with Hoechst (blue); (**F**) Quantification of the percentage of EdU positive cells/total cells; (G) CCK-8 assay was used to estimate total cell number. Statistical results were indicated as mean ± SEM, *n* = 3. * *p* < 0.05. vs. NC.

**Figure 5 ijms-20-01816-f005:**
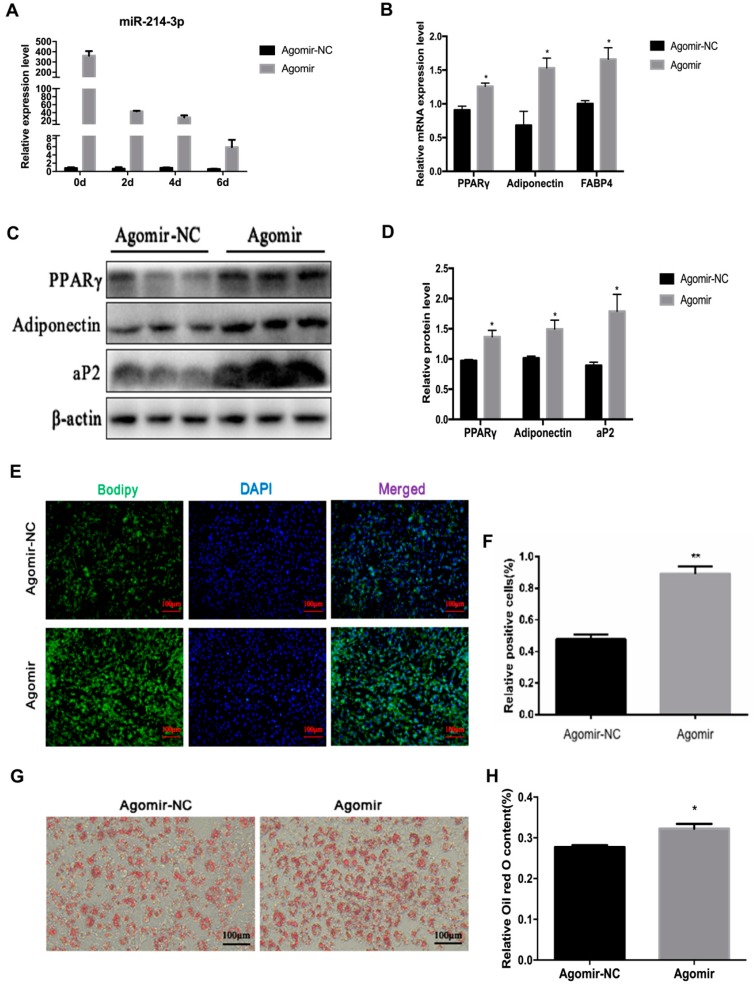
MiR-214-3p enhances adipocytes differentiation in 3T3-L1 cells. MiR-214-3p agomir or negative control (NC) were transfected into cells at 80% density at 50 nM. (**A**) The overexpression efficiency of miR-214-3p after transfection with the miR-214-3p agomir; (**B**) RT-qPCR was used to detect adipogenesis genes, PPARγ, FAS, Adiponectin, and FABP4 on the sixth day after induction; (**C**) Western blot analysis of adipogenesis genes on the sixth day of differentiation; (**D**) Quantification of Western blot analysis of PPARγ, Adiponectin, aP2, (**E**) BODIPY staining was performed in cells on the sixth day of differentiation, the lipid droplets of the cells were stained with BODIPY (green) and the total nuclei were stained with 4′,6-diamidino-2-phenylindole (DAPI) (blue), after transfection with the miR-214-3p agomir; (**F**) BODIPY positive cells/DAPI cells percentage was quantified, (bar size = 100 μm); (**G**) Oil Red O staining, differentiated 3T3-L1 cells on the sixth day were stained with Oil Red O; (**H**) Triglycerides content was measured by spectrophotometric analysis at 490 nm. Statistical results are indicated as mean ± SEM, *n* = 3. * *p* < 0.05; ** *p* < 0.01, vs. NC.

**Figure 6 ijms-20-01816-f006:**
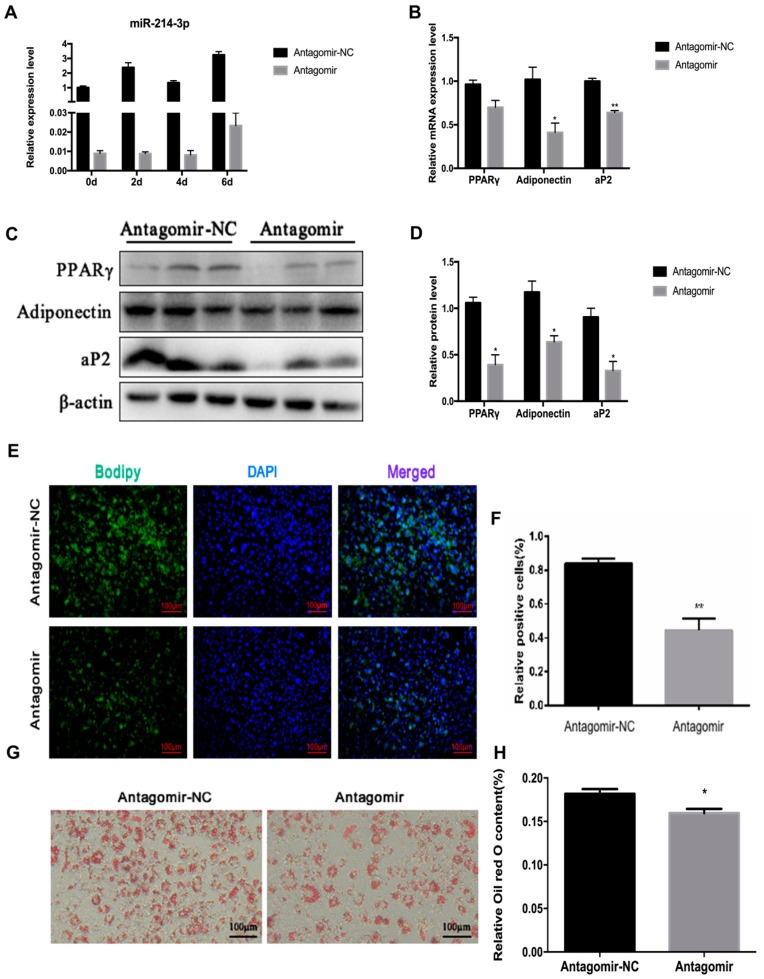
MiR-214-3p inhibitor suppresses adipogenic differentiation. MiR-214-3p antagomir or negative control (NC) were transfected into cells at 80% density at 50 nM. (**A**)The knockdown efficiency of miR-214-3p after transfection with the miR-214-3p antagomir; (**B**) RT-qPCR was used to detect mRNA levels of PPARγ, FABP4, and Adiponectin on the sixth day of differentiation after transfection with the miR-214-3p antagomir; (**C**) Western blot analysis of adipogenesis genes on the sixth day of differentiation after transfection with miR-214-3p antagomir; (**D**) Quantification of Western blot analysis of PPARγ, Adiponectin, aP2; (**E**) BODIPY staining was performed in cells on the sixth day of differentiation, the lipid droplets of the cells were stained with BODIPY (green) and the total nuclei were stained with DAPI (blue), after transfection with the miR-214-3p agomir; (**F**) Quantification of the percentage of BODIPY positive cells/DAPI cells; (**G**) Oil Red O staining, differentiated 3T3-L1 cells were stained with Oil Red O; (**H**) Triglyceride content was measured by spectrophotometric analysis. Data are presented as mean ± SEM. *n* = 3. * *p* < 0.05; ** *p* < 0.01, vs NC.

**Figure 7 ijms-20-01816-f007:**
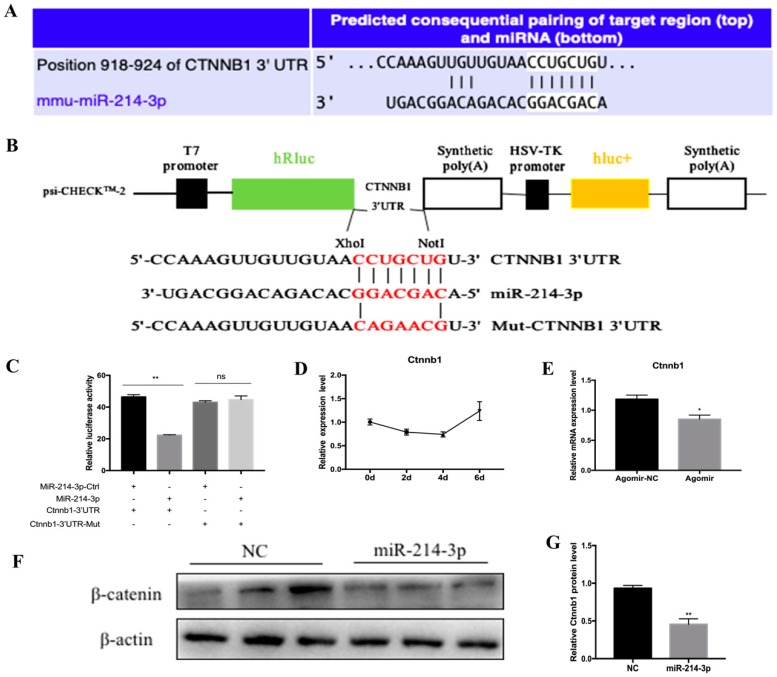
MiR-214-3p can target Ctnnb1 during 3T3-L1 preadipocyte differentiation. (**A**) Ctnnb1 is a direct target gene of miR-214-3p; (**B**) Schematic diagram of the dual luciferase reporter psi-CHECK 2.0-Ctnnb1 3’UTR; (**C**) Dual luciferase assay was performed by co-transfection of miR-214-3p agomir and wild-type vectors or mutant vectors. Relative luciferase activity was represented by Renilla Luciferase/Firefly Luciferase (RLUC/FLUC); (**D**) Expression profile of Ctnnb1 during adipogenic differentiation; (**E**) The relative Ctnnb1 mRNA expression levels after treatment with the miR-214-3p agomir; (**F**) Western blot analysis of Ctnnb1 protein expression after transfection with the miR-214-3p agomir at six days of adipogenic differentiation; (**G**) The quantification of β-catenin protein levels. Results are represented as mean ± SEM. *n* = 3. * *p* < 0.05; ** *p* < 0.01, vs. NC.

**Table 1 ijms-20-01816-t001:** Sequences primers.

Gene Name	Forward (5′-3′)	Reverse (5′-3′)
*PPARγ*	CGCTGATGCACTGCCTATGA	AGAGGTCCACAGAGCTGATTCC
*aP2*	CGATCCCAATGAGCAAGTGG	TGGGTCAAGCAACTCTGGAT
*Adiponectin*	GGCAGGAAAGGAGAACCTGG	AGCCTTGTCCTTCTTGAAGAG
*CDK4*	AGTTTCTAAGCGGCCTGGAT	AACTTCAGGAGCTCGGTACC
*Cyclin B*	AACTTCAGCCTGGGTCG	CAGGGAGTCTTCACTGTAGGA
*Cyclin E*	GCTTGCTCCGGGGATGAAAT	GCGAGGACACCATAAGGAAATCTG
*Cyclin D*	TAGGCCCTCAGCCTCACTC	CCACCCCTGGGATAAAGCAC
*Ctnnb1*	TCCCATCCACGCAGTTTGAC	TCCTCATCGTTTAGCAGTTTTGT
*β-actin*	GTCCCTGACCCTCCCAAAAG	GCTGCCTCAACACCTCAACCC

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
