# Peer review of "MicroRNA-214-3p Targeting Ctnnb1 Promotes 3T3-L1 Preadipocyte Differentiation by Interfering with the Wnt/β-Catenin Signaling Pathway"

_ijms, 2019, doi:10.3390/ijms20081816_

Reviewer 1 Report

In the manuscript entitled “MicroRNA-214-3p targeting Ctnnb1 promotes 3T3-L1 preadipocyte differentiation through Wnt/β-catenin signaling pathway” Fengxue Xi and colleagues demonstrate that miR-214 has a role in preadipocyte proliferation and differentiation and that the mechanism of action is mediated by Ctnnb1.

In the first section of the paper the authors show that miR-214 is expressed in white and brown adipose tissue. However a relative expression (vs muscle ) is shown. This has no sense. An absolute quantification is more appropriate. Line 74-75 “real‐time fluorescence quantification showed that miR‐214‐3p had a  higher level (of what???) of expression in the adipose tissue of mice” need rephrasing.

Next the authors show that miR-214 was increased in the adipose tissue of HFD mice. This point has been recently shown in a paper by Linyun He et al (Obesity associated miR-199a/214 cluster inhibits adipose browning via 2 PRDM16-PGC-1α transcriptional network, Diabetes 2018) where a positive correlation between miR-199a/214 cluster expression and obesity has been shown both in mice and humans.

To confirm a function of miR‐214‐3p in adipogenesis the authors analyze the expression of this miRNA in 3T3‐L1 adipogenesis. They observed “a trend of increasing first and then decreasing during proliferation and differentiation”. This is the first point in contrast with the He et al. paper showing a clear decrease of miR-214 during white and brown adipocyte differentiation. Please comment this point.

To investigate the role of miR‐214‐3p in adipogenesis the authors perform gain and loss of function experiments checking the effect of miRNA modulation on preadipocyte proliferation and differentiation. To overexpress miR-214 an Agomir has been transfected in 3T3‐L1 cells. After transfection the miR-214 level is 500 folds higher than in the control cells. Is this a physiological level? We have some doubts about. Besides this large miRNA amount, the effect on cyclin expression were very modest. As far as concern mRNA expression only 1 of the four cyclins tested show significant increase. Western blot experiments are also shown but a quantification of the signals in relation to the internal standards are absolutely necessary , particularly because of the variability among the experiments. The loss of function approach has the same problems. Overall these results are in contrast with He et al., paper that, in line with the observation of a decrease of miR-199a and miR-214 in adipose tissue differentiation, show that miR-199a/214 suppresses brown adipocyte differentiation and beige fat development by directly targeting Prdm16 and PGC-1α, two key 14 transcriptional regulators of adipose browning. I think that this is a crucial point that the authors should afford in the discussion while they mention the roles of miR-214 in osteosarcoma or C2C12 myoblast differentiation whose relation with the focus of this study is questionable.

Moreover in the manuscript there are many language and conceptual problems such as the sentence (lane 264) “   miRNAs function by transcriptional inhibition or translational degradation of target genes….”

Author Response

Dear Reviewer,

Thank you for reading our manuscript entitled “MicroRNA-214-3p targeting Ctnnb1 promotes 3T3-L1 preadipocyte differentiation by interfering with Wnt/β-catenin signaling pathway” (Manuscript ID: ijms-475780) and reviewing it. Those comments are all valuable and very helpful for revising and improving our paper, as well as the important guiding significance to our researches. We have studied comments carefully and have made correction which we hope meet with approval. Revised portion are marked in red in the paper. The main corrections in the paper and the responds to your comments was uploaded in Word format, please check it out.

Reviewer 2 Report

Comments to Authors.

In this work, the Authors reported the involvement of miR‐214‐3p in the regulation of the preadipocyte growth and differentiation, thus extending our knowledge in the physiological miRNA action.

My Concerns

1. As in the manuscript the Authors wrote “Taken together, these data showed that miR‐214‐3p promotes adipocyte differentiation through directly targeting Ctnnb1 and inhibiting the Wnt/β‐catenin signaling pathway”, it would be more appropriate to change the title (for istance: “MicroRNA-214-3p targeting Ctnnb1 promotes 3T3-L1 preadipocyte differentiation interfering with Wnt/β-catenin signaling pathway”.

2.  Figure 1 panel A understanding is not immediate…it would be better to report the miR-214-3p expression with respect to the liver or spleen.

3. Comparing the panels B and E in Figure 1 the increase in body weight in the groups (panel B) is no very high, but the significance is p<0,001. On the contrary, the increase of miR-214-3p expression in iWAT is very high, but the significance is p<0,01. These effects may depend on the reproducibility of the data. Can the Authors confirm them?

4. In addition, the figure legend of panel E should respect the order of samples indicated in the graph.

5. In Figure 2 panels B and C the Authors should indicate the cell event analyzed to allow an immediate understanding of the Figure.

6. The sentence in page 4 lines 97-99 “ We further….”should be re-written because data reported in Figure 2 panel D are referred to cell proliferation and differentiation, not to pre-adypocite specifically.

7. In material and methods the Authors should better define agomiR and antagomir. In addition the Authors should indicate in the figure legends the transfection time of these constructs when are used.

8. In Figure 3 panel B is not clear the significance of data reported. In panel D the Authors should increase the magnitude of the pictures to allow a better understanding of the data. For the same reason the Authors should add the densitometry analysis of the protein bands reported in Figure 3C.

9. In Figure 4 there are the same concerns indicated in Figure 3.

10. In Figure 5 the Authors should modify the magnitude of pictures in panel D and add the densitometry in panel C.

11. In Figure 6 panel B the Authors should delete p=0.05. The Authors could report this data in discussion paragraph. The Authors should modify the magnitude of pictures in panel D and add the densitometry in panel C.

12. In Figure Legend 6 what it means “(F)……. and statistical analysis of Ctnnb1 protein expression at 6d of adipogenic differentiation”?. And “(G) The quantification of β‐catenin protein level? In addition, the mention to panel G is not reported in the main text.

13.  Although the manuscript is generally well-written and easy to read, some sentence could be modified (i.e in the sentence page 4-5 lines 111-114, page 7 lines 150-152 and similar “respectively” could be deleted; while the sentence in pag 9 lines 191-193 is not clear…I think that “and” should be eliminated). In addition, in page 13 line 275 the Authors could change CTNNB1 with Ctnnb1.

Author Response

Dear Reviewer,

Thank you for reading our manuscript entitled “MicroRNA-214-3p targeting Ctnnb1 promotes 3T3-L1 preadipocyte differentiation by interfering with Wnt/β-catenin signaling pathway” (Manuscript ID: ijms-475780) and reviewing it. Those comments are all valuable and very helpful for revising and improving our paper, as well as the important guiding significance to our researches. We have studied comments carefully and have made correction which we hope meet with approval. Revised portion are marked in red in the paper. The main corrections in the paper and the responds to your comments was uploaded in PDF format, please check it out.

Round  2

Reviewer 1 Report

I have no further comments or suggestions. Only one point :

miRNA usually results in the inhibition of protein production due to mRNA

degradation or translational repression. They do not affect protein degradation . Please correct the sentence at lanes 295-296.

Author Response

Dear Reviewer,

We are very grateful to your comments and suggestions for our manuscript “MicroRNA-214-3p targeting Ctnnb1 promotes 3T3-L1 preadipocyte differentiation by interfering with Wnt/β-catenin signaling pathway” (Manuscript ID: ijms-475780)”. We have checked the manuscript and revised it according to your comments. The correction in the paper and the responds to your comments are as following:

Point: I have no further comments or suggestions. Only one point:

miRNA usually results in the inhibition of protein production due to mRNA degradation or translational repression. They do not affect protein degradation. Please correct the sentence at lanes 295-296.

Response: Thank you very much for your suggestions. We have revised it according to your comments:

“miRNAs function by binding to the 3'UTR of the target genes, resulting in transcriptional inhibition or translational degradation of the target genes.” has been rewritten as “miRNAs bind to the 3’UTR of target mRNAs to induce mRNA degradation or translational repression”. Seeing in line 356-357.

Once again, thank you very much for your comments and suggestions.

Yours sincerely,

Dr. Taiyong Yu